# Cognitive–Behavioral Treatment of Obsessive–Compulsive Disorder: The Results of a Naturalistic Outcomes Study

**DOI:** 10.3390/jcm11102762

**Published:** 2022-05-13

**Authors:** Andrea Gragnani, Vittoria Zaccari, Giuseppe Femia, Valerio Pellegrini, Katia Tenore, Stefania Fadda, Olga Ines Luppino, Barbara Basile, Teresa Cosentino, Claudia Perdighe, Giuseppe Romano, Angelo Maria Saliani, Francesco Mancini

**Affiliations:** 1Associazione Scuola di Psicoterapia Cognitiva (APC-SPC), 00185 Rome, Italy; femia@apc.it (G.F.); valerio.pellegrini@uniroma1.it (V.P.); tenore@apc.it (K.T.); fadda@apc.it (S.F.); dottoressaluppino@gmail.com (O.I.L.); basile@apc.it (B.B.); cosentino@apc.it (T.C.); perdighe@apc.it (C.P.); romano@apc.it (G.R.); saliani@apc.it (A.M.S.); mancini@apc.it (F.M.); 2Department of Human Sciences, Marconi University, 00193 Rome, Italy; 3Department of Social and Developmental Psychology Sapienza, University of Rome, 00185 Rome, Italy

**Keywords:** obsessive–compulsive disorder, cognitive–behavioral therapy, cognitive interventions, outcomes, effectiveness, naturalistic study

## Abstract

Cognitive–behavioral therapy is a well-established treatment for obsessive–compulsive disorder (OCD). There are a variety of cognitive and behavioral strategies, and it is necessary to analyze the outcomes of the treatments. The aim of the present study is to verify the effectiveness of a treatment that combines evidence-based procedures and specific cognitive interventions highlighting the issue of acceptance. Forty patients with OCD were recruited and underwent a specific treatment procedure. All patients had a psychodiagnostic assessment for OCD using the Y–BOCS (Yale–Brown obsessive–compulsive scale) performed twice: before treatment (t0) and after nine months (t1). Data analysis showed a decrease in the scores between t0 and t1 according to the Y–BOCS in terms of the interference, severity, and impairment of obsessive–compulsive symptoms. A repeated-measures ANOVA showed a significant reduction in symptoms after treatment, with values of *F* (1, 39) = 137.56, *p* < 0.001, and *η^2^* = 0.78. The ANOVA results were corroborated by a Wilcoxon signed-rank test. A reliable change index analysis indicated that 33 participants reported improvements in symptoms, of which 23 were clinically significant. The results showed clinical relevance for OCD treatment and highlighted how this cognitive procedure favored positive outcomes.

## 1. Introduction

Obsessive–compulsive disorder (OCD) is a common clinical mental condition with an estimated lifetime prevalence of 2.3% [1,2,3].

According to the Diagnostic and Statistical Manual of Mental Disorders [4], OCD is characterized by obsessions (“*recurrent and persistent thoughts, urges, or impulses that are experienced at some time during the disturbance as intrusive and unwanted, and that in most individuals cause marked anxiety or distress”)* and by compulsions *(“repetitive behaviors or mental acts, that the individual feels driven to perform in response to an obsession or according to rules that must be applied rigidly. The behaviors or mental acts are aimed at preventing or reducing anxiety or distress, or preventing some dreaded event or situation”)* [4].

The quality of life, productivity, and functioning of patients with OCD are often significantly compromised [5], and this often leads to the chronicity of the disorder [6]. This is all the more serious considering the long duration of the disease: a person who develops the disorder at a young age has a high probability of suffering for a long time and seeing their existential fulfillment severely compromised [7,8,9]. Several studies have documented that a longer duration of untreated disease leads to worse outcomes and prognoses [10]. Therefore, it is critically important that patients with OCD receive appropriate treatment in a timely manner to reduce suffering and improve functioning. This increasingly raises the importance of providing effective treatment for OCD [11] and developing research in this direction.

The international guidelines [12], in relation to the degree of symptomatic impairment, recommend pharmacological therapy and cognitive–behavioral therapy (CBT), the treatments currently proven to be most effective. CBT refers to exposure and response prevention (ERP), with or without the inclusion of cognitive therapy strategies, and it is the psychological therapy of first choice for OCD; patients are taught to confront and tolerate conditions that provoke obsessions and compulsions and resist acting on them [13]. Additionally, the guidelines [12] suggest providing OCD-specific cognitive therapy (CT) for patients who refuse ERP. The 2005 NICE guidelines [12], which were based on a meta-analysis of existing trial data, advocate the use of low-intensity psychological treatments (including ERP) for adult patients with mild symptoms of OCD, pharmacological therapy (SSRI) for patients with moderate symptoms or patients with mild illness who cannot tolerate low-intensity psychological treatment, and combination therapy (SSRI and CBT with ERP) for patients with more severe or resistant illness.

The prognosis of patients with OCD has improved significantly since the late 1960s, i.e., when the first cases of patients treated with ERP were reported; the ERP procedure involves exposure to the feared stimuli and the interruption of behaviors usually implemented [3,14]. Patients treated with the ERP procedure not only showed a marked improvement in symptoms but also appeared stable in follow-ups after the conclusion of the treatment.

In fact, in the last twenty years, numerous meta-analyses and systematic reviews have been conducted to evaluate the effectiveness of CBT with ERP in treating OCD through several randomized controlled trials (RCTs) [15,16,17]. The effect sizes of CBT in OCD are among the largest in the psychotherapy literature [18]. Nevertheless, a substantial proportion of patients do not attain remission, and drop-out rates have been documented [19,20,21,22]. ERP has often been described as a challenging treatment because it involves confronting anxiety-provoking cues, and it has been estimated that between 25 and 30% of patients refuse the ERP treatment [20,21] and around 30% leave the treatment [21,22].

However, some authors [16] have highlighted, in authoritative systematic reviews and meta-analyses, that the reasons for patients declining to participate, refusing ERP, or dropping out are not specified in the method sections of the RCTs. Indeed, these data led researchers, in several literature reviews, to highlight the methodological criticalities that call for a review of the evidence supporting the efficacy of CBT involving ERP [16,23,24].

In a systematic review and meta-analysis, Öst et al. [16] underline the importance, in comparing the studies that investigated the efficacy of CBT treatments, of the differences in the tools used to measure the change caused by the therapy; the Y–BOCS (Yale–Brown obsessive–compulsive scale) interview is currently the gold standard in the evaluation of OCD. Furthermore, it seems that a significant proportion of patients refuse to undergo ERP because it is perceived as a frightening treatment, reaffirming the importance of investigating the issue of the refusal rate of CBT with ERP and the rate of premature drop-out. The results of this meta-analysis [16] show that CBT appears to be more effective than other treatments, such as psychopharmacological treatments, waiting lists, and placebos. The study confirms the effectiveness of ERP for treating OCD, with or without additional elements of CT. The choice of an ERP treatment that uses modern CT strategies or that combines both of these elements does not affect the effectiveness in terms of the improvement measured with the Y–BOCS.

In relation to the refusal rate of 25% that has often been reported for ERP [19], the results from this review show that studies vary considerably with respect to the number of patients declining participation. The results in the present meta-analysis show high variability in drop-out rates across studies, with a mean attrition rate of 15%. Cognitive therapy had the lowest drop-out rate with 11.4%, ERP had a rate of 19.1%, and the combination of ERP/CBT and antidepressants had the highest rate with 32.0%. On average, 15% of patients refused the offer of treatment, with a range from 0% (e.g., in [25]) to 63% [26].

Another meta-analysis conducted by Leeuwerik et al. [27] found refusal and drop-out rates of 15.6% and 15.9%, respectively, suggesting that over 30% of eligible patients who are recommended CBT for OCD fail to initiate or complete treatment. This review did not find a significant difference in drop-out rates for different types of CBT (i.e., ERP, CBT, and CT), which is consistent with other meta-analyses [16,28,29].

It can be concluded that, although effective pharmacological and psychotherapeutic treatments have been developed in recent years, as supported by authoritative systematic reviews and meta-analysis [15,16,17,23,24,27], some problems remain unsolved:
(a)a significant percentage of participants do not respond to treatment;(b)many participants are unwilling or do not tolerate ERP.

To date, there are no reliable data that highlight the problems that lead patients to reject ERP and abandon treatment. However, it has previously been documented that ERP is known as a challenging treatment that involves confronting anxiety-inducing stimuli.

To overcome these limitations, it is necessary to use specific intervention procedures that can facilitate the implementation of ERP and, consequently, extend the efficacy of the therapy. Therefore, it seems important to understand how ERP refusal rates can be reduced and how to effectively export these findings to clinical practice, since RCTs are often not very ecological or generalizable in clinical practice. It can be inferred that if the enrolled participants refuse ERP, the effectiveness of the treatment is reduced.

To fill the gaps, when considering the efficacy of CBT ERP for treating OCD, it is important to verify and analyze whether specific cognitive interventions can increase adherence to the treatment and increase the efficacy of the therapy.

The main aim of the present study is to verify the effectiveness of a treatment that combines evidence-based procedures (a CBT–ERP intervention procedure developed according to international guidelines) and specific cognitive interventions highlighting the issue of acceptance, where the rationale is to facilitate the ERP intervention and favor the acceptance process as a cognitive restructuring [30,31,32]. The study was conducted in a sample of patients with OCD through a nine-month outcome assessment and included a comparison with the state of symptoms detected in the first psychodiagnostic evaluation carried out before the start of treatment.

The acceptance process, based on assuming and accepting a higher level of risk as well as adopting less prudential attitudes and reasoning, appeared essential in our clinical practice for increasing the effectiveness of CBT intervention as it favors ERP. In previous research [30], we verified the usefulness of a specific cognitive intervention, performed before ERP, aimed at helping the patient to tolerate exposure to the feared critical stimuli by renouncing safety-seeking behaviors, particularly compulsions. Subsequently, ERP intervention was conceptualized, presented to the patient, and managed as a practical exercise in the acceptance of exposure to stimuli of increasing intensity and the progressive reduction in safety-seeking behaviors, including compulsions. In this way, we obtained a very low rate of refusal of therapy and drop-out, about 8%, and we saw significant clinical improvements in about 80% of patients. The same procedure, with similar results, was used by Zaccari et al. [32]. The intent that guided us was also to verify the results obtained in the 2006 study and to determine whether our intervention procedure reduces the refusal of ERP and the drop-out rate while maintaining the effectiveness of the CBT/ERP intervention, replicating a naturalistic design [30].

A limitation, in fact, of RCTs on CBT is the possible lack of representativeness of the treatment conditions [18] and, therefore, the decreased generalizability of the results to clinical settings [33]. Another limitation is the application of very strict inclusion and exclusion criteria in RCTs regarding comorbidity, concurrent medications, age range, and symptom profiles (types of obsessions and compulsions). Furthermore, the randomization of treatment conditions may prevent patients from participating in RCTs, making the treated samples unrepresentative of the population of treatment-seeking patients with OCD [34]. In RCTs, a wide variation in outcome measures has been applied, and the differences in outcome measures represent a challenge when comparing studies. A comparison of multiple outcome measures may bias the calculated effect sizes because the standard deviations (SDs) may vary substantially between measures [35]. Therefore, it is clear how necessary studies replicating clinical conditions are; moreover, additional evidence from effectiveness studies focusing on more naturalistic conditions is urgently needed to identify which effects can be attained in everyday clinical care, as such information is less documented in the literature and more useful for indications in clinical practice [36,37].

## 2. Materials and Methods

### 2.1. Participants

In the present study, we aimed to test the effectiveness of the treatment through parametric and non-parametric analyses. Specifically, we decided to implement a repeated-measures ANOVA (analysis of variance) and a Wilcoxon signed-rank test. Therefore, to determine the sample size, we conducted a distinct power analysis using G*power for both types of analysis we were interested in performing. Based on the results outlined by previous literature (e.g., [15,16,17]), we opted for the expected medium effect size for both power analyses. Specifically, regarding the repeated-measures ANOVA, we set a value of f equal to 0.25 along with an error probability of 0.05, a conventional power threshold of 0.80, and a correlation among repeated-measures of 0.30. The analysis revealed a minimum sample size of 46 participants. As for the Wilcoxon signed-rank test, we opted for similar parameters (*d_z_* = 0.5, *α* = 0.05, and 1 *− β* = 0.80). In this case, the analysis yielded a minimum sample size of 35 participants in order to achieve a statistical power of 0.80.

Therefore, we recruited a sample of 43 Italian adult patients with OCD (male: 25; age: M = 32.70; SD = 8.91) who required CBT treatment for OCD at the Center for Cognitive Psychotherapy in Rome from 2020 to 2021. The data presented in this work concern a group of unselected patients diagnosed with OCD, consecutively recruited according to the treatment request criterion.

The inclusion criteria for the study were: being over 18 years of age; having a primary diagnosis of OCD; having undergone a psychodiagnostic assessment with the Y–BOCS before undertaking CBT; having undertaken CBT following the psychodiagnostic evaluation; comorbidities with other clinical profiles were excluded when reported in the exclusion criteria.

The exclusion criteria included: significant mental comorbid disorders (psychosis, borderline personality, mania, alcohol or drug abuse or dependence, and impaired cognitive function); a current marked risk to self (self-harm or suicide); and dissociation symptoms (assessed with a clinical interview and psychodiagnostic assessment).

Of the 43 participants recruited, 40 started and continued the treatment; two patients interrupted it early, one in the third session (first phase of treatment) who declared that he was not available to maintain a weekly psychotherapy commitment, and the other in the fifth session (second phase of treatment) without providing an explanation even when requested. They did not resume therapy sessions. One patient refused ERP in the eighteenth session (fourth phase of treatment).

Participant characteristics, gender, age, comorbidity, medications, and OCD subtypes for each participant are shown in Table 1, Table 2 and Table 3.

### 2.2. Procedure

Before undertaking the treatment, all the participants provided informed consent for the tracing and disclosure of data for scientific purposes. A diagnosis of OCD was made according to DSM-5 criteria [4] based on an extensive clinical examination and psychodiagnostic assessment by an expert clinician using the structured clinical interview for diagnosis (SCID-5-CV) [38]. In addition, during the diagnostic interview, information was collected on the duration of the disease, comorbidities, and previous and/or current pharmacological treatments. All patients had a psychodiagnostic assessment for OCD through the Y–BOCS performed twice: before treatment (t0) and after nine months (t1). Evaluation of the outcomes was carried out through the Y–BOCS. All patients underwent a specific treatment procedure. A psychodiagnostic assessment was performed after 9 months of treatment. The procedure was approved by the Ethical Committee of the School of Cognitive Psychotherapy (Italy).

### 2.3. Measure

The Yale–Brown obsessive–compulsive scale (Y–BOCS) [39,40] is a widely used clinician-administered interview for assessing the presence and severity of OCD symptoms in adults and is considered a gold-standard instrument to measure the severity of OCD. The Y–BOCS includes a symptom checklist, a clinician-rated checklist designed to guide a structured interview to determine the target symptoms for treatment, and a severity scale (with ten items) that assesses the severity of obsessions (five items) and compulsions (five items) using a five-point Likert-type scale (from 0 to 4; the total score of the scale may range between 0 and 40). A total score of ≥16 is considered to be indicative of clinically significant OCD. The scales showed strong internal consistency for both the total score and each subscale (α = 0.83) [41]. In the present study, we found good reliability for the total score on the Y–BOCS (α = 0.86), as well as for the obsession (α = 0.73) and compulsion (α = 0.88) sub-dimensions at t0. Similarly, reliability was also good also at t1 (total score: α = 0.89; obsession: α = 0.89; compulsion: α = 0.88). In this study, the total score on the Y–BOCS and the subscale scores (obsessions and compulsions) are used.

### 2.4. Treatment

All participants underwent CBT and a specific treatment procedure [30,31,32] according to international guidelines [12] that recommend psychopharmacological therapy and CBT that involves the use of ERP [13], with or without the inclusion of cognitive therapy strategies; this is similar to the most effective treatments.

Our specific treatment procedure and CBT put ERP at the center of the intervention; treatment is implemented taking into account the problems documented in the literature and observable in clinical practice: the refusal of ERP and drop-outs. The intervention procedure is divided into five phases and follows a preliminary assessment of the patient (see Appendix A for a summary of OCD treatment). The mean number of psychotherapy sessions is 32 one-hour sessions (range 28–36). The number of treatment sessions, in particular ERP, is in line with international guidelines [12].

Each phase may require more or fewer sessions according to the patient’s needs.

The first phase consists of reconstruction and the sharing of the functioning scheme (diagrammatic model for understanding the disorder [42] (see Appendix A)) of the patient’s disorder and specific symptomatology. The purpose behind this phase is threefold. First, it allows for a rational planning of the psychotherapeutic intervention. Second, it facilitates awareness of obsessive functioning and promotes distancing from pathogenic beliefs. Third, it conveys the sense and functioning of the disorder to the patient, thus increasing the possibility that the patient feels well understood by the therapist and improving the therapeutic alliance. The functioning scheme is rebuilt following the ABC model [42] (see Appendix A—a checker patient example).

The second phase involves the attempt to modulate the beliefs that support the negative or threat evaluation of the critical event (for example, having brushed up against someone on the street. The event may be perceived, remembered, or only hypothesized) and that sustain the motivation. Among the main cognitive restructuring techniques, the probability pie and cumulative probability methods are used to reduce the perceived probabilities of events [43]; methods to reduce responsibility include the defense lawyer technique, the responsibility pie chart, and the courtroom technique [44,45,46]; there are also interventions for the normalization of forbidden thoughts [22,47,48]. However, these interventions are not always sufficient on their own to counter the typical tendency of obsessive patients in search of absolute certainty that the feared threat will not happen.

In the third phase, the strategy of the intervention is reversed, and an attempt is made to help the patient enter the order of ideas that threat, contamination, the not just right experience (NJRE), doubt, and uncertainty can be accepted. This phase of the intervention is the original part of our work because it has been specifically designed to reduce the refusal of ERP treatment and drop-outs by facilitating the acceptance of gradual exposure to feared stimuli and the progressive renunciation of compulsions.

Accepting a threat reduces investments in prevention. The reduction in protective investment also involves cognitive processes that are less prudentially oriented and, therefore, yield a less dramatic representation of the threat [31,49,50].

For example, accepting the risk of having left a gas tap open implies a lower focus on the possibility of a gas leak and an explosion and a greater focus on reassuring possibilities, such as having closed the tap securely and that the safety valves are activated. Furthermore, it also involves information processing that is less corroboratory and more refutatory than focusing on negative possibilities [51], thus providing reassuring information. This is achieved through the modification of four different patient assumptions: belief in the power to reduce threat, contamination, NJRE, doubt and uncertainty; that it is advantageous to invest in this direction; that this is morally necessary; and that, if not adhered to, suffering will increase, last a very long time, and create obstacles to normal daily activities [49]. It is clear that one is more willing to accept a threat without attempting to take action if one realizes that the compulsions and safety-seeking behaviors are ineffective or even counterproductive.

Therefore, it is useful to understand that the certainty of neutralizing the feared threat is impossible and that, therefore, some degree of threat is inevitable. This can be achieved by helping patients to focus on how much and in how many ways they have tried to neutralize the threat and the fact that the threat has always reappeared and will always do so, no matter how much they invest in this direction. It may be useful to help patients understand that their attempts to neutralize the threat with certainty are useless and that the threat is inevitable. It is useful to point out to patients the ephemeral nature of the safety that they sometimes feel they have obtained through compulsions.

For example, this may be accomplished by pointing out to patients that the tranquility that they have achieved after numerous checks is not guaranteed at all because nothing excludes with certainty the possibility that they made a mistake during the last check. Behavioral experiments are [52,53,54] useful to help patients understand that compulsions and safety-seeking behaviors, for example, the suppression of blasphemous thoughts, have a paradoxical effect.

Obsessive patients, when exposed to the feared stimulus, tend to consider only the costs, in terms of risk and suffering, that they pay if they do not implement compulsions and safety-seeking behaviors; they do not consider the costs of the compulsions or the safety-seeking behaviors. It is clear that exposing oneself to what is feared without adhering to compulsions or safety-seeking behaviors has a cost and, therefore, it is difficult to pursue. However, it can become acceptable if this cost is compared with that of the compulsions and safety-seeking behaviors and if the latter is more significant than the former. To help the patient clarify the two sets of costs and compare them, we found the “two chairs” technique useful [49,55,56,57] (see Appendix A). For obsessive patients, not adhering to any compulsions or safety-seeking behaviors in response to a feared threat often implies a sense of guilt that hinders acceptance. For example, in the case of patients who, as soon as they leave their apartment, are assailed by the doubt that they have left the gas tap open, suppose it is clear to them that this is the usual exaggerated obsessive doubt and that a gas leak appears quite improbable. Likewise, let us imagine that they also realize that going back to check involves a serious delay in getting to work and, therefore, considerable sanctions. Despite all this, they are driven to go back, and they control themselves with the fear of being guilty of neglecting a possibility of which they were aware.

Patients can be helped to reduce the sense of guilt with the double standard technique [44], in particular, its modified version [49,56,58] (see Appendix A). If the intervention has a good result, patients can be asked to train themselves to repeat it when exposed to the threatening stimulus and before implementing the usual compulsions and safety-seeking behaviors. If it does not work, an attempt can be made, with due caution, to help patients reduce the moral motivation, leading them to compare the risks of the guilt they assume if they renounce the compulsions and safety-seeking behaviors when faced with the threat with the responsibility for the damage that fueling their obsessive disorder certainly causes, both for themselves and others. Substantially, the goal is to realize that undertaking ERP is morally more correct than avoiding it.

Another obstacle to accepting exposur and renouncing compulsions and safety-seeking is the idea that suffering will last forever and indeed increase to become unbearable. This point can be addressed with psychoeducation, that is, by explaining to patients that, in reality, the discomfort one accepts by not carrying out the rituals tends to reduce spontaneously and that, above all, subsequent exposures will be easier. To consolidate psychoeducation, small ERP experiments can be used to give up the attempts of solution. It is appropriate to conceptualize the ERP as a series of practical exercises in accepting gradually increasing threat levels [59,60].

In the fourth phase, ERP is implemented [22]. It is important to highlight that ERP is conceptualized as a series of practical exercises in accepting gradually increasing threat levels and gradually giving up compulsions [59,60]. The procedure involves the combined application of two components: exposure and response prevention. In this study, ERP was carried out in vivo following the canonical indications [22]; therefore, we do not describe the procedure.

The fifth phase includes action to reduce vulnerability to OCD and the general disposition of patients with OCD to a fear of guilt [61,62], according to an important research tradition [44,61,63,64,65,66,67,68] that supports the role of guilt and the fear of guilt in the genesis and maintenance of OCD. In fact, several studies suggest that in obsessive patients, there are often memories of early experiences characterized by the fear of parental reprimand, by hyper-responsibility, or by an education very attentive to morality. These are experiences, therefore, that plausibly sensitized the future patient to the fear of guilt [69,70,71,72].

The intervention addresses the historical vulnerability of a patient by working on elements of the patient’s life history that favored the onset of the disorder, such as experiences, sensitizing episodes, and factors predisposing the patient to the specific disorder/problem. With imaging rescripting [73,74], it is possible to intervene effectively in the memories of these experiences, helping the patient to re-elaborate them in a less dramatic and more functional way. This phase’s purpose is to make sensitizing experiences less relevant from an emotional point of view [70,71,72]. Relapses are always possible, but they can have a very different outcome if the patient experiences them without being frightened by them or dramatizing them and while having clear indications on how to manage them and when to contact the psychotherapist [75].

## 3. Results

### 3.1. Repeated-Measures ANOVA

We aimed to investigate the effectiveness of a specific CBT intervention procedure in the treatment of obsessive–compulsive disorder. To this end, we first implemented a repeated-measures ANOVA that considered the total Y–BOCS score shown by patients before and after being exposed to the CBT intervention procedure. As expected, the analysis revealed a significant effect of the treatment: *F* (1, 39) = 137.56, *p* < 0.001, and *η^2^* = 0.78. Participants showed a considerable reduction in OCD-related symptoms between the two measurements (Figure 1). Specifically, from an average score of 28.03 (*SD* = 5.03) at baseline, the Y–BOCS score decreased to 14.95 (*SD* = 6.58) after exposure to treatment. The difference between the two means turned out to be marked: *MeanDiff* = 13.08, *SE* = 1.12, *p* < 0.001, and 95% CI = 10.82, 15.33. To gauge the practical magnitude of the effect, we computed *Cohen’s d_rm, pooled_* [76] for the differences among the mean scores for t0 and t1. This effect size estimation considers the pooled standard deviation, controlling for the intercorrelation of both groups (i.e., 0.31). We thus found a noticeable effect size associated with the treatment (*d_rm, pooled_* = 1.58, 95%CI = 1.03, 2.13). This result gave us an initial consistent indication of the effectiveness of the specific CBT treatment procedure.

### 3.2. Wilcoxon Signed-Rank Test

Due to the relatively small number of participants, we tried to add corroboration to the ANOVA results by means of a non-parametric test. Such a non-parametric analysis was also intended to enrich our investigation by examining potential differences in treatment efficacy in distinct symptomatic sub-dimensions of OCD. Thus, we implemented a Wilcoxon signed-rank test for the non-parametric comparison of the total score of Y–BOCS, and of the related obsession and compulsion sub-dimensions, across pre- and post-treatment measurements. Descriptive statistics associated with these measures are reported in Table 4. As can be seen in Table 5, the results were consistent with those of the repeated-measures ANOVA. Specifically, the analysis showed a decrease in the total score on the Y–BOCS that pertained to the entire sample of patients (*z* = 5.51, *p* < 0.001). Furthermore, the analyses also highlighted a significant effect of the treatment on 39 of the 40 patients in relation to the obsessive dimension (*z* = 5.49, *p* < 0.001). Only one patient showed a higher post-treatment score in respect of the baseline. Similarly, the analysis underlined the effectiveness of the treatment for 39 of the 40 patients for the compulsion dimension (*z* = 5.45, *p* < 0.001). In this case, one patient showed the same score across the pre- and post-treatment measurements. Finally, all test statistics were associated with an effect size (*r*). These effect sizes were computed by dividing the Z test statistic by the square root of the total number of observations [77]. Thus, we found an *r* of 0.61 (95%CI = 0.37, 0.78) for both test statistics associated with the Y–BOCS total score and the sub-dimensions. As for the sub-dimension of compulsion, the analysis indicated an *r* of 0.60 (95%CI = 0.36, 0.77).

### 3.3. Reliable Change Index and Clinical Significance

As a further integration of the parametric and non-parametric analyses, we also computed the reliable change index (RCI) and clinical significance [78]. Reliability data for the RCI were obtained from Melli et al. [41]. The *clinsig* R package [79] was employed to conduct this analysis. It allowed us to examine OCD-related symptom score variations across the measurements (i.e., pre- and post-treatment) for each patient. The RCI and clinical significance were computed for the total Y–BOCS score in order to assess the potential symptom improvement of patients by comparing their mean score at the baseline to the mean score after the treatment. For each participant, a reliable change thus corresponded to a score reduction of two standard deviations with respect to the specific Y–BOCS score he/she exhibited pre-treatment. Clinical significance, on the other hand, was obtained when patients reported a score post-treatment that was two standard deviations lower than the overall mean of the sample at baseline (i.e., criterion A [78]). In the present sample, this criterion corresponded to a score equal to 17, which also matched with the generally accepted clinical cut-off for the Y–BOCS (reference). As shown in Figure 2, analysis provided corroboration of the effectiveness of the proposed specific CBT treatment procedure. Consistent with the ANOVA and Wilcoxon signed-rank test, we found that the overall mean score post-treatment was lower than criterion A (i.e., 17). Furthermore, analysis also indicated that 33 of the 40 patients reported a reliable decrease in OCD-related symptoms. In other words, these 33 patients showed a Y–BOCS score at their post-treatment measurement that was two standard deviations lower than their individual pre-treatment score. Finally, analysis revealed that 23 of these 33 patients reported symptom improvements that could be considered clinically significant. That is, 23 patients exhibited a Y–BOCS score after the treatment that was lower than criterion A.

## 4. Discussion

The purpose of this study was to verify and analyze the effectiveness of a treatment that combines evidence-based procedures (CBT–ERP according to international guidelines) and specific cognitive interventions to facilitate the implementation of ERP [30]. The aim was to increase adherence to the treatment and, consequently, extend the efficacy of the therapy in a sample of patients with OCD through a nine-month outcome assessment and a comparison of the outcomes with the symptomatic state detected in the first psychodiagnostic evaluation carried out before the start of treatment. Our goal was to focus attention on certain problems that remain unsolved in the literature and in clinical practice, namely the significant percentage of patients with OCD who do not respond to treatment and the large number of participants who are not willing to or do not tolerate ERP. Indeed, the scientific literature documents that a significant proportion of patients refuse to undergo ERP, which is perceived as a frightening treatment, reaffirming the importance of investigating both the refusal rate of CBT and the rate of premature drop-out.

Therefore, it is important to underline and continue research focusing on aspects crucial for evaluating outcomes and effectiveness, such as the refusal of ERP, the drop-out rate, and the appropriate measurement of outcomes [16]. In our study, the participants showed a considerable reduction in OCD-related symptoms between the two measurements, and we were able to observe a significant decrease in the Y–BOCS scores at t1 in terms of the interference, severity, and impairment of the obsessive–compulsive symptoms. We thus found a noticeable effect size associated with the treatment. This result indicates the effectiveness of the treatment.

These results were also corroborated by a non-parametric analysis that gave us a more precise indication of how many patients with OCD actually improved. We verified that, in all patients, there was a significant improvement in obsessive–compulsive symptoms. In addition, we observed with respect to the sub-dimensions (obsessions and compulsions of the Y–BOCS) that one patient had a t1 score equal to t0 for obsessions and one patient had a t1 score worse than at t0 in regard to compulsions; however, all improved at t1 in terms of their Y–BOCS total scores.

It is noteworthy that the substantial improvements highlighted by the normalization of the Y–BOCS scores are more than those normally obtained in controlled and non-naturalistic studies: only two of the treated patients did not obtain normalization of symptoms in the obsession and compulsion sub-dimensions in the post-treatment score compared to the baseline; these two patients did, however, improve in terms of general symptoms. Even reliable change index analysis indicated that 33 of the 40 patients reported a reliable decrease in OCD-related symptoms, of which 33 were clinically significant (i.e., two standard deviations below the baseline mean).

Our results highlight a global improvement in symptomatology. These results were obtained using Y–BOCS, which is considered the gold standard for OCD assessment. In a systematic review and meta-analysis, Öst et al. [16] documented that previous studies that investigated the efficacy of CBT treatments had differences in the tools used to measure the change due to the therapy and recommended the use of the Y–BOCS in OCD evaluation. With respect to our results, the interesting data concern adherence to treatment and drop-out rates. Of the 43 participants enrolled, 3 out of 43 patients interrupted treatment (about 7%), two of which (4,5%) interrupted treatment early with non-evaluable benefits; only one refused ERP (2,3%). This result confirmed our hypothesis and was in line with previous research [30] that obtained a very low rate of refusal of therapy and drop-out (about 8%) and achieved clinically significant improvements in about 80% of patients.

In addition, our results have great clinical relevance in addressing certain problems, including the fact that a substantial proportion of patients do not attain remission, that 25–30% of patients refuse ERP treatment [20,21], and that around 30% leave treatment [19,21,22]; our results highlight how such problems could be solved by favoring the ERP acceptance process. Although the results are not from a controlled study, they document a very low rate of refusal of therapy and drop-out, unlike what has been widely observed in previous meta-analyses that reported a high level of variability with refusal and drop-out rates of 15.6% and 15.9% respectively, suggesting that over 30% of eligible patients who are recommended CBT for treatment of OCD fail to initiate or complete treatment [27]. This drop-out rate is consistent with two earlier meta-analyses of studies evaluating CBT for treating OCD [16,29].

Indeed, from our results, it can be deduced that participants treated with this specific intervention procedure did not refuse ERP; therefore, this supports its effectiveness. In fact, it is interesting to focus on the specific cognitive intervention procedures that emphasize the issue of acceptance to facilitate the implementation of ERP in order to expand the effectiveness of CBT with ERP and act on the problem of outcomes.

Stricter cognitive interventions, in which the emphasis is on changing the cognitive structures associated with the role of responsibility, the fear of guilt, the overestimation and acceptance of risks, and the not just right experience (NJRE) [44,53,61,63,65,66,67,68,80,81,82,83,84,85], have proven their effectiveness.

Over the last decade or so, acceptance has taken on great importance as a process, a strategy, and a technique. There has been a proliferation and propagation of therapeutic approaches that are, more or less explicitly, based on acceptance. In any event, acceptance as a way of dealing with life’s troubles and frustrations is not, strictly speaking, a new theme. Long before Hayes and Kabat-Zinn introduced acceptance commitment therapy (ACT) [86] and mindfulness [87], respectively, some of the world’s major religions (such as Christianity, Islam, and Buddhism), a number of important currents of philosophy (such as Stoicism and the Eastern philosophical tradition), and even cognitive therapy, in particular rational emotive therapy (RET) [42], pointed to acceptance as a coping strategy [88,89].

We have, indeed, expanded and developed a specific procedure that goes beyond the experiential acceptance contemplated by the interventions of ACT [86], mindfulness [87], or interventions aimed at promoting an intentional and conscious way of distancing from one’s mental contents; our approach is aimed at changing cognitive processes, assuming and accepting a higher level of risk, and adopting less prudential attitudes and reasoning. This acceptance process, in our clinical experience, and as can be seen from the results obtained, increases the effectiveness of CBT as it favors ERP and reduces the number of drop-outs.

Indeed, it seems to us that the advantages of our treatment [30,31] that combines a procedure of proven efficacy with interventions that emphasize the acceptance of specific emotional and cognitive states, compared to treatment with ERP alone, demonstrate that our treatment has greater applicability to and collaboration with the exposure treatment and yields more stable results and a greater number of patients who achieve normalization of symptoms. Therefore, we believe that the theme of acceptance is a fundamental ingredient in preparing for ERP.

We believe that acceptance, as a process, moves through a standard cognitive restructuring procedure and towards a higher level of risk acceptance with regard to the possibility of eliminating the risk of the threat, the possibility of engaging in risk reduction, the duty to reduce risk, and the possibility of investments in strategies other than threat reduction [50,55,90,91]. The third phase of treatment, in our opinion, seems to promote the fundamental condition of ERP, decreasing the possibility of dropping out due to fear that is often generated during the phases of factual exposure.

This places importance on the acceptance process in therapy for OCD. The acceptance process can be considered a crucial step because it favors ERP and limits the refusal of ERP and drop-out; therefore, its use is recommended.

Finally, it is important to reflect on the limitations of RCTs, both due to a possible lack of representativeness of treatment conditions [18] and because of the very strict inclusion and exclusion criteria regarding comorbidity, concurrent medications, or subtype OCD profiles. The prospect of randomization into treatment conditions may prevent patients from participating in RCTs, making the treated sample unrepresentative of the population with OCD [34], unlike naturalistic studies. This study, although it used a naturalistic design, made use of a structured treatment procedure in which each phase had a specific rationale for its intervention. This represents a strong level of control, as all patients followed a structured treatment. The intervention that we have proposed, and in particular the work on acceptance, seems to have been effective on a sample made up of patients with a primary diagnosis of OCD, including different subtypes and with other comorbidities. This indicates that the proposed intervention aimed at accepting threats is also effective for all OCD subtypes and for OCD profiles in the presence of other comorbidities. The results obtained seem to have great clinical relevance given the clinical characteristics of the sample; unlike non-naturalistic studies in which participants are selected on the basis of symptoms and, therefore, on specific inclusion/exclusion criteria, our study placed a strong emphasis on real clinical practice.

Therefore, although it is fundamental for scientific research to conduct RCTs, it is also important to pay attention to naturalistic outcomes to identify which effects can be attained in everyday clinical care; such considerations are less documented in the literature and more useful for indications in clinical practice [36,37] and, therefore, for the generalizability of the results to clinical settings [33].

Although our study provides interesting results, it is important to bear in mind some limitations. Firstly, our sample sizes for analyses were small; secondly, we did not evaluate the presence of possible variables that could have positively moderated the outcomes.

Significant improvement in symptoms may have been moderated by other variables. Therefore, it would be desirable for future research studies to also use other outcome measures that detect anxiety, mood, self-criticism, rumination, or worry processes.

Furthermore, for future research, it would be advisable to compare classical CBT with ERP to our CBT treatment with its specific cognitive interventions of acceptance and ERP, in order to verify whether there is a real positive therapeutic outcome and a lower drop-out rate while also making a comparison with the most recent literature data.

## 5. Conclusions

In this study, we wanted to verify the previous results obtained in a study of patients with OCD [30] and determine whether our intervention procedure that combines evidence-based procedures and specific cognitive interventions highlighting the issue of acceptance would reduce the refusal of ERP and the drop-out rate while maintaining the effectiveness of the CBT/ERP intervention, all while replicating a naturalistic design. The results show a significant improvement in obsessive–compulsive symptoms, good adherence to treatment, a low drop-out rate, and a noticeable effect size associated with the treatment.

Despite some limitations, these results may provide an interesting starting point for further clinical studies. Our results show clinical relevance for OCD treatment and highlight that the CBT procedure treatment used [30] favors an improvement in the symptoms and, therefore, a positive outcome. This underlines the importance of the specific intervention procedure for those seeking to replicate the results obtained from empirical research, and it is useful for clinicians in regard to giving clear and effective indications for the treatment of OCD.

## Figures and Tables

**Figure 1 jcm-11-02762-f001:**
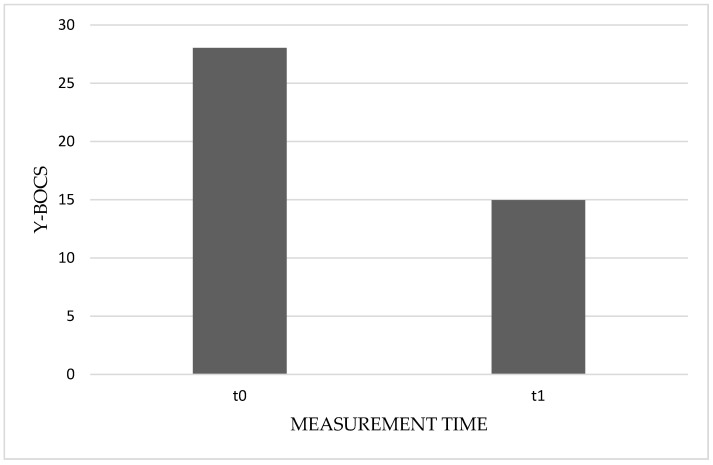
Average score of the Yale–Brown obsessive–compulsive scale (Y–BOCS) before (t0) and after (t1) CBT specific procedure.

**Figure 2 jcm-11-02762-f002:**
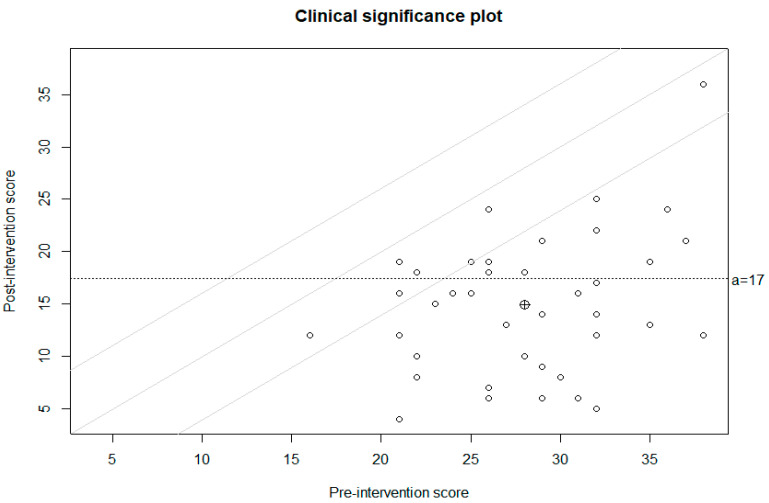
Reliable change index and clinical significance.

**Table 1 jcm-11-02762-t001:** Descriptive statistics of the whole sample.

Variables	*Frequency*	*Percentage*
**Gender**		
Male	25	58.1
Female	18	41.9
**Nationality**		
Italian	42	97.7
Other	1	2.3
**Medications**		
Yes	17	39.5
No	26	60.5
	** *Mean* **	** *SD* **
**Age**	32.70	8.91
**Disorder Duration in Months**	93.42	100.14

*Note*: N = 43.

**Table 2 jcm-11-02762-t002:** Distribution of distinct comorbidities.

**Comorbidity 1**	** *Frequency* **	** *Percentage* **
APD	2	4.7
BD-II	1	2.3
BIP 2	1	2.3
BN	1	2.3
BPD	3	7.0
BPD (Tr)	2	4.7
DEP	7	16.3
DPD (Tr)	3	7.0
IAD	1	2.3
NPD	1	2.3
NPD (Tr)	4	9.3
OCPD (Tr)	2	4.7
PAN	4	9.3
PPD	1	2.3
PPD (Tr)	1	2.3
SAD	1	2.3
NONE	8	18.6
**Comorbidity 2**	** *Frequency* **	** *Percentage* **
BPD	1	2.3
DEP	2	4.7
DPD	1	2.3
NPD (Tr)	1	2.3
NPD and DEP	1	2.3
OCPD	2	4.7
PPD	1	2.3
PPD (Tr)	3	7.0
SAD	3	7.0
UPD	3	7.0
NONE	25	58.1

*Note:* N = 43; Comorbidity: DEP = Depression; IAD = Illness anxiety disorder; BD-II = Bipolar disorder; PAN = Panic disorder; BN = Bulimia nervosa; SAD = Social anxiety disorder; APD = Avoidance personality disorder; BPD = Borderline personality disorder; PPD = Paranoid personality disorder; NPD = Narcissistic personality disorder; DPD = Dependent personality disorder; OCPD = Obsessive–compulsive personality disorder; UPD = Unspecified personality disorder; (Tr) = Traits of personality disorders.

**Table 3 jcm-11-02762-t003:** Distribution of OCD subtypes.

**OCD Subtype 1**	** *Frequency* **	** *Percentage* **
AS	2	4.7%
C and W	9	20.9%
CH	14	32.6%
U	17	39.5%
Washer	1	2.3%
**OCD Subtype 2**	** *Frequency* **	** *Percentage* **
C and W	5	11.6%
CH	10	23.3%
U	6	14.0%
None	22	51.2%

*Note:* N = 43; OCD Subtypes: CH = Checking; U = Unacceptable taboo thoughts; C and W = Contamination and washing; VO = Various obsessions; AS = All subtypes.

**Table 4 jcm-11-02762-t004:** Descriptive statistics. Means, standard deviations, median and inter-quartile range (IQR) of the total score of the Yale–Brown obsessive–compulsive scale (Y–BOCS) and of the related obsession (OBS) and compulsion subdimensions (COM), before (t0) and after (t1) the treatment.

*Measures*	*Time*	*Mean*	*SD*	*Median*	*IQR*
Y–BOCS	t0	28.03	5.30	28.00	8
	t1	14.95	6.58	15.50	9
OBS	t0	14.65	2.45	14.50	4
	t1	8.13	3.84	8.50	5
COM	t0	13.38	3.50	14.00	4
	t1	6.83	3.79	7.00	5

*Note:* Y–BOCS: Yale–Brown obsessive–compulsive scale; OBS: Obsession; COM: Compulsion.

**Table 5 jcm-11-02762-t005:** Wilcoxon signed-rank test.

Measures	Negative Ranks	Positive Ranks	Ties	Total	z	*p*
Y–BOCS	40	0	0	40	−5.51	<0.001
OBS	39	1	0	40	−5.49	<0.001
COM	39	0	1	40	−5.45	<0.001

*Note:* Y–BOCS: Yale–Brown obsessive–compulsive scale; OBS: Obsession; COM: Compulsion.

## Data Availability

Not applicable.

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
