# Peer review of "Cognitive–Behavioral Treatment of Obsessive–Compulsive Disorder: The Results of a Naturalistic Outcomes Study"

_jcm, 2022, doi:10.3390/jcm11102762_

Round 1
Reviewer 1 Report
This is an interesting article and piece of software. The article is well written.
Title and abstract: The title is appropriate for the content of the article. The abstract is concise and accurately summarizes the essential information of the paper although it would be better if the authors define more precisely the the Y-BOCS 16(Yale-Brown Obsessive-Compulsive Scale).
The clinical case presentation is comprehensive and detailed but there are some minor points that should be clarified the figures.
The authors need to show more clearly what is original and useful in their findings ‐ what it is, exactly, that will be important to other scholars in the field.
I hope my comments make sense ‐ please do not hesitate to contact me if not.
Author Response
Response to Reviewer 1 Comments
Point 1: This is an interesting article and piece of software. The article is well written.
Title and abstract: The title is appropriate for the content of the article. The abstract is concise and accurately summarizes the essential information of the paper although it would be better if the authors define more precisely the the Y-BOCS (Yale-Brown Obsessive-Compulsive Scale).
Response 1: Thank you for the appreciation you made of our work. We feel that your suggestions really contributed to improve the quality of our manuscript.
We have made several changes in the all body of the manuscript. Therefore, we proceeded to make a new upload of the main document (look at the new file that has been uploaded with the changes).
According to point 1, the Y-BOCS is explained in detail in the method section (measure). It was not specified in the abstract as there was a maximum of words allowed.
Point 2: The clinical case presentation is comprehensive and detailed but there are some minor points that should be clarified the figures.
Response 2: This comment is not clear so we ask you for explanations. We did not describe a clinical case in the paper. Are you most likely referring to the treatment procedure?
Point 3: The authors need to show more clearly what is original and useful in their findings ‐ what it is, exactly, that will be important to other scholars in the field.
Response 3: Thank you for focusing on our results. This part you can see in the lines 540-553 and in section conclusion.
Furthermore, thanks to your suggestion, we have added another part (see lines 554 - 556) to emphasize our results and in particular the treatment.
Reviewer 2 Report
This is an interesting study which seeks to improve the efficacy of ERP for OCD by adding CBT-based strategies to enhance adherence to the intervention. The authors not that while ERP is very effective, participants may disengage early due to fear of exposure, and that strategies are required to enhance adherence.
Although the definition of CBT can be muddy, it was my understanding that this therapy is widely understood to come under the ‘umbrella’ of CBT approaches. The introduction would therefore be more interpretable if the authors provided a description early on of the differences between ERP alone and ERP-CBT.
Please conclude the background section with a clear description of the specific aims of this study.
Please include a box with a line-by-line summary of phase content/number of sessions in brief, as this section of the manuscript is quite lengthy. For service-planning purposes, authors should also consider explaining how this compares to the average duration of ERP only.
Given that this study aims to investigate drop-out rates, it would also be pertinent to consider refusal rates – how many potentially eligible participants were approached but did not consent to participate?
Please clarify ‘interrupted’ treatment. Was treatment paused and then resumed? If so, for how long and why? How were their data treated in the analysis? Why was imputation of missing data for these participants not considered?
The use of acceptance to describe both treatment non-refusal as well as acceptance of the risk of feared outcomes can be difficult to follow at times.
The authors present some justification for use of a naturalistic approach (to better replicate the everyday clinical environment) as compared to an RCT. I agree that such studies can provide evidence that complements more rigorous and controlled studies. However, as the authors note evidence from a meta-analysis that there is a high level of variability in drop-out rates across studies, it would be important to put the present findings in context with the minimum and maximum dropout rates observed in prior ERP-only studies, as well as symptom severity and medication status. This is particularly important as a participant screening and recruitment flowchart was not provided. Including these numbers would also support the author’s claim that naturalistic studies are more representatitve of treatment-seeking populations.
In the limitations section, could the authors elaborate on what possible outcomes could have influenced symptomatology to guide future research on what variables should be measured and accounted for?
Author Response
Response to Reviewer 2 Comments
Point 1: This is an interesting study which seeks to improve the efficacy of ERP for OCD by adding CBT-based strategies to enhance adherence to the intervention. The authors not that while ERP is very effective, participants may disengage early due to fear of exposure, and that strategies are required to enhance adherence.
Response 1: Thank you for accepting the review of our work. Your suggestions have been fundamental to review and to reflect better on its limitation and improve the quality of our manuscript. We have made several changes in the all body of the manuscript. Therefore, we proceeded to make a new upload of the main document (look at the new file that has been uploaded with the changes).
According to point 1, this part is well expressed throughout the work as it represents the rational.
Point 2: Although the definition of CBT can be muddy, it was my understanding that this therapy is widely understood to come under the ‘umbrella’ of CBT approaches. The introduction would therefore be more interpretable if the authors provided a description early on of the differences between ERP alone and ERP-CBT.
Response 2: We decided not to focus on the differences between ERP alone and ERP-CBT as, according to international guidelines and the scientific community, CBT encompasses ERP (cognitive and behavioral techniques).
The intervention procedure we have proposed is consistent with international guidelines (NICE, 2005) which recommend pharmacological therapy and Cognitive-Behavioral Therapy (CBT), the most effective treatments, provided in relation to the degree of symptomatic impairment. CBT involves the use of the Exposure and Response technique (ERP; Kozak & Foa, 1997) with the inclusion of cognitive therapy strategies. In addition, the guidelines (NICE, 2005) suggest providing OCD-specific cognitive therapy for patients who reject ERP. With our procedure we have placed more emphasis on cognitive techniques in order to favor behavioral exposures (ERP).
Point 3: Please conclude the background section with a clear description of the specific aims of this study.
Response 3: Our aim is expressed at lines 123-129/141-144: “The aim of the present study is to verify the effectiveness of a treatment that combines evidence-based procedures (CBT – ERP intervention procedure according to international guidelines) and specific cognitive interventions highlighting the issue of acceptance, where the rationale is to facilitate the ERP intervention and favor the acceptance process as a cognitive restructuring [30, 31, 32], in a sample of OCD patients through a nine-month outcome assessment and a comparison with the state of symptoms detected in the first psychodiagnostic evaluation carried out, before the start of treatment”.
“The intent that guided us was also to verify the results obtained in the 2006 study (reference 30) and whether our intervention procedure reduces the refusal of ERP and drop-out rates while maintaining the effectiveness of the CBT/ERP intervention, replicating a naturalistic design”
How can they be further clarified? Thank you for providing us with this suggestion.
Point 4: Please include a box with a line-by-line summary of phase content/number of sessions in brief, as this section of the manuscript is quite lengthy. For service-planning purposes, authors should also consider explaining how this compares to the average duration of ERP only.
Response 4: Thank you for providing us with this valuable information. We have drawn up the table that summarizes the treatment: phases, contents and number of sessions. The average duration of the intervention was shown in the table in the appendix section and in the body of the text (see lines 239 - 241).
Our treatment does not compare with a treatment that only involves ERP. ERP is included in CBT treatment according to the international guidelines for OCD (NICE 2005).
We have proceeded to insert in the manuscript that the number of treatment sessions, in particular ERP, are in line with international guidelines (NICE 2005: see lines 241).
Point 5: Given that this study aims to investigate drop-out rates, it would also be pertinent to consider refusal rates – how many potentially eligible participants were approached but did not consent to participate?
Response 5: No patient opposed the treatment. All subjects who arrived at our clinical center (Center for Cognitive Psychotherapy in Rome, a clinical center known in Italy for being specialized CBT for OCD) requested psychotherapy for OCD in the period 2020 - 2021 (see lines 181 – 182).
Point 6: Please clarify ‘interrupted’ treatment. Was treatment paused and then resumed? If so, for how long and why? How were their data treated in the analysis? Why was imputation of missing data for these participants not considered?
Response 6: Of the 43 subjects recruited, 40 started and continued the treatment; two patients interrupted it early with non-evaluable benefits; one refused the ERP.
"Interreputed" means that they voluntarily finished the psychotherapy and did not resume the sessions (see line 195). Only one patient refused to carry out ERP.
These three subjects were not considered in the statistical analysis as there was no outcome measure (t1).
Point 7: The use of acceptance to describe both treatment non-refusal as well as acceptance of the risk of feared outcomes can be difficult to follow at times.
Response 7: Thank you for raising this issue. Risk acceptance refers to the process of accepting exposing yourself to the feared threat of obsessions to obtain ERP treatment. Acceptance instead used as non-acceptance of ERP treatment means that patients do not tolerate or do not want to undertake ERP and therefore the risk of acceptance of the feared threat through exposure exercises.
We tried to change and replace the term acceptance when we were talking about “treatment non-refusal (ERP) as well as acceptance of the risk of feared” to make it clearer to the reader.
We made change at lines:
- 109: b) a high number of subjects are unwilling or do not tolerate ERP.
- 132 - 135: In previous research [30] we verified the usefulness of a specific cognitive intervention, before ERP, aimed at helping the patient to expose to the feared critical stimuli by renouncing safety-seeking behaviors, particularly compulsions.
- 300 - 301: It is clear that exposing oneself to what is feared without taking compulsions and safety seeking behaviors has a cost and therefore it is difficult to pursue.
- 324 - 325: Substantially, the goal is to realize that undertake ERP is morally more correct than avoiding it.
- 326: Another obstacle to implementation of to deal with exposure, renouncing the compulsions and safety-seeking behaviors is the idea that suffering will last forever and indeed will increase, becoming unbearable.
- 460-461: a large number of subjects who are not willing to or do not tolerate ERP.
Point 8: The authors present some justification for use of a naturalistic approach (to better replicate the everyday clinical environment) as compared to an RCT. I agree that such studies can provide evidence that complements more rigorous and controlled studies. However, as the authors note evidence from a meta-analysis that there is a high level of variability in drop-out rates across studies, it would be important to put the present findings in context with the minimum and maximum dropout rates observed in prior ERP-only studies, as well as symptom severity and medication status. This is particularly important as a participant screening and recruitment flowchart was not provided.
Including these numbers would also support the author’s claim that naturalistic studies are more representatitve of treatment-seeking populations.
Response 8:
Thank you for giving us this important indication. We inserted the following sentences at lines:
- 495-497. “Of the 43 subjects enrolled, 3 out of 43 patients interrupted treatment (about 7%) of which two (4,5%) interrupted treatment early with non-evaluable benefits and only one refused ERP (2,3%)”.
- 503-509. “Although the results are not from a controlled study they document a very low rate of refusal of therapy and drop-out unlike what has been widely observed in previous
meta-analysis in which it is reported a high level of variability in refusal and dropout rates of 15.6% and 15.9% respectively, suggesting that over 30% of eligible patients who are recommended CBT for OCD fail to initiate or complete treatment [27]. This dropout rate is consistent with two other earlier meta-analyses of studies evaluating CBT for OCD [16, 29]”.
These average percentages can be found in the meta-analysis. There is no reference numerical range.
Point 9: In the limitations section, could the authors elaborate on what possible outcomes could have influenced symptomatology to guide future research on what variables should be measured and accounted for?
Response 9: Significant improvement in symptoms may have been moderated by other variables. Therefore, it would be desirable for future research studies to also use other outcome measures that detect anxiety, mood, self-criticism, rumination or worry processes.
(see lines 579– 581).
